# Population Pharmacokinetic Modeling and Pharmacodynamic Target Attainment Simulation of Piperacillin/Tazobactam for Dosing Optimization in Late Elderly Patients with Pneumonia

**DOI:** 10.3390/antibiotics9030113

**Published:** 2020-03-06

**Authors:** Noriyuki Ishihara, Nobuhiro Nishimura, Kazuro Ikawa, Fumi Karino, Kiyotaka Miura, Hiroki Tamaki, Takahisa Yano, Takeshi Isobe, Norifumi Morikawa, Kohji Naora

**Affiliations:** 1Department of Pharmacy, Shimane University Hospital, 89-1 Enya-cho, Izumo-shi, Shimane 693-8501, Japan; 2Faculty of Pharmacy, International University of Health and Welfare, 137-1 Enokizu, Ookawa-shi, Fukuoka 831-8501, Japan; 3Department of Clinical Pharmacotherapy, Hiroshima University, 1-2-3 Kasumi, Minami-ku, Hiroshima-shi, Hiroshima 734-8553, Japan; 4Department of Internal Medicine, Division of Medical Oncology and Respiratory Medicine, Shimane University Faculty of Medicine, 89-1 Enya-cho, Izumo-shi, Shimane 693-8501, Japan

**Keywords:** piperacillin, tazobactam, late elderly patients, pneumonia, population pharmacokinetic/pharmacodynamic analysis

## Abstract

The aim of this study was to develop a population pharmacokinetic model for piperacillin (PIPC)/tazobactam (TAZ) in late elderly patients with pneumonia and to optimize the administration planning by applying pharmacokinetic/pharmacodynamic (PK/PD) criteria. PIPC/TAZ (total dose of 2.25 or 4.5 g) was infused intravenously three times daily to Japanese patients over 75 years old. The plasma concentrations of PIPC and TAZ were determined using high-performance liquid chromatography and modeled using the NONMEM program. PK/PD analysis with a random simulation was conducted using the final population PK model to estimate the probability of target attainment (PTA) profiles for various PIPC/TAZ-regimen–minimum-inhibitory-concentration (MIC) combinations. The PTAs for PIPC and TAZ were determined as the fraction that achieved at least 50% free time > MIC and area under the free-plasma-concentration–time curve over 24 h ≥ 96 μg h/mL, respectively. A total of 18 cases, the mean age of which was 86.5 ± 6.0 (75–101) years, were investigated. The plasma-concentration–time profiles of PIPC and TAZ were characterized by a two-compartment model. The parameter estimates for the final model, namely the total clearance, central distribution volume, peripheral distribution volume, and intercompartmental clearance, were 4.58 + 0.061 × (CL_cr_ − 37.4) L/h, 5.39 L, 6.96 L, and 20.7 L/h for PIPC, and 5.00 + 0.059 × (CL_cr_ − 37.4) L/h, 6.29 L, 7.73 L, and 24.0 L/h for TAZ, respectively, where CL_cr_ is the creatinine clearance. PK/PD analysis using the final model showed that in drug-resistant strains with a MIC > 8 μg/mL, 4.5 g of PIPC/TAZ every 6 h was required, even for the patients with a CL_cr_ of 50–60 mL/min. The population PK model developed in this study, together with MIC value, can be useful for optimizing the PIPC/TAZ dosage in the over-75-year-old patients, when they are administered PIPC/TAZ. Therefore, the findings of present study may contribute to improving the efficacy and safety of the administration of PIPC/TAZ therapy in late elderly patients with pneumonia.

## 1. Introduction

Piperacillin/tazobactam (PIPC/TAZ, 8:1) is a β-lactam/β-lactamase inhibitor combination with in vitro activity against a broad spectrum of aerobic and anaerobic Gram-positive and Gram-negative bacteria, including penicillinase-, cephalosporinase-, and extended-spectrum β-lactamase-producing bacteria. In many guidelines for infectious diseases, including pneumonia, PIPC/TAZ, like fourth-generation cephems, carbapenems and new quinolones, is rated as a first-choice drug for a variety of patients at risk for resistant bacteria [1,2]. Particularly for severe cases of pneumonia, the use of broad-spectrum antimicrobials such as PIPC/TAZ is recommended [3,4,5].

The third most common cause of death in the world is lower respiratory infections [6]. Moreover, in Japan, increased mortality from pneumonia suggests that elderly patients and those with underlying diseases or deteriorated general conditions are recommended antimicrobial chemotherapy, which is effective against drug-resistant bacteria [7]. Today, in many cases, the dosage regimen for the elderly is adjusted based on the package insert or books such as the Sanford Guide [8]. However, this dosage regimen recommended in such a way does not take into account bacterial susceptibility.

Evidence-based antimicrobial chemotherapy is required to prevent the emergence of drug-resistant infectious microorganisms caused by the unintended administration of low-dose antibiotics and to limit the spread of healthcare-associated and community-acquired drug-resistant strains.

Random simulation-based pharmacokinetic (PK)/pharmacodynamic (PD) software applications are known to be useful tools for providing individualized antimicrobial chemotherapy in accordance with the PK/PD theory of antimicrobials [9]. Establishing a population PK model is necessary in planning an optimized therapeutic regimen. Population PK models on PIPC/TAZ have been reported in young adult patients with nosocomial infections [10,11], sepsis [12,13], febrile neutropenia [14], acute infections [15], and obesity [16], and those undergoing continuous renal replacement therapy [17]. Additionally, antimicrobial chemotherapy with a combination of a β-lactam/β-lactamase inhibitor requires that there is a sufficient amount of the inhibitor to inactivate β-lactamases, which allows the β-lactam to exhibit its antibacterial activity [18]. However, there are very few reports on population PK analysis and PK/PD analysis for PIPC and TAZ in late elderly patients [19], and the optimal dosing method considering both PIPC and TAZ simultaneously has not been established.

The aim of this study is to develop a population PK model for PIPC/TAZ in late elderly patients with pneumonia and to present optimized dosing regimens by applying PK/PD criteria.

## 2. Results

### 2.1. Patient Characteristics

The patient backgrounds are shown in Table 1. A total of 18 cases were investigated, with 14 males and 4 females. The mean age was 86.5 ± 6.0 years old. The mean body weight and body mass index were 45.5 kg and 19.1, respectively. The mean creatinine clearance (CL_cr_) value calculated with the Cockcroft–Gault equation was 38.0 ± 11.1 mL/min.

### 2.2. Population PK Model

#### 2.2.1. PIPC

A total of 100 plasma PIPC concentration data (Figure 1A) were used for the population PK modeling. As Akaike information criterion (AIC) values indicated that the two-compartment model (AIC, 1040.069) described the data better than the one-compartment model (AIC, 1063.038), the two-compartment model was chosen as the base model. Therefore, the PK parameters were clearance (CL), volume of distribution of the central compartment (V_c_), intercompartmental clearance (Q), and volume of distribution of the peripheral compartment (V_p_). The mean parameters of the base model were: CL = 4.57 L/h; V_c_ = 5.30 L; Q = 21.8 L/h; V_p_ = 7.01 L. The interindividual variability for V_p_ was finally fixed as zero, because it was negligibly small (η < 0.00173).

In order to build the covariate model by the forward inclusion process, the linear, exponential, and allometric models were examined. As a result, a significant difference from the base model was detected only in the several linear models. Among the linear models, the fixed-effects parameter with the largest Δobjective function (ΔOBJ) change (−5) was used as the final model, although CL_cr_, age, and body weight each also significantly affected CL_cr_. As age and body weight each showed a high correlation with CL_cr_, they were not additionally incorporated into CL to avoid a collinearity effect. None of the examined covariates (body weight, body mass index, total protein, serum albumin, serum creatinine, or CL_cr_) significantly affected V_c_, Q, and V_p_. The parameter estimates for the final model are shown in Table 2. All parameter estimates were in the range of the 95% CI using the bootstrap method.

#### 2.2.2. TAZ

A total of 100 plasma TAZ concentration data (Figure 1B) were employed for the population PK modeling. Because the AIC value in the two-compartment model (596.955) was smaller than that in the one-compartment model (615.059), the two-compartment model was chosen as the base model. The inter-individual variability for V_p_ was negligibly small (η < 0.00459), so it was finally fixed as zero. The mean parameters of the base model were: CL = 5.00 L/h; V_c_ = 6.21 L; Q = 25.8 L/h; V_p_ = 7.82 L.

During the forward inclusion process to build the covariate model, no significant difference from the base model was found either in the linear, exponential, or allometric models. TAZ, which has similar pharmacokinetics to PIPC, is known to be excreted primarily from the kidney, most of which is eliminated as unchanged in urine. Therefore, similarly to PIPC, the linear model with the minimal *p*-value was selected as the fixed-effects parameter for TAZ. None of the examined covariates significantly affected V_c_, Q, or V_p_. The parameter estimates for the final model are shown in Table 3. All parameter estimates were in the range of the 95% CI using the bootstrap method.

#### 2.2.3. Model Evaluation

The goodness-of-fit plots are depicted in Figure 2A–D. As observed in the plots with observed vs. population and individual predicted concentrations, the data are uniformly scattered around the line of unity.

### 2.3. PK/PD Analysis

In PIPC, several typical patient populations (CL_cr_ = 10, 20, 30, 40, 50, and 60 mL/min) were supposed based on the final population PK model. The mean protein binding of PIPC and TAZ is 30% [20], and this was used to simulate unbound concentrations. Changes in probability of target attainments (PTAs) of various PIPC dosing regimens with CL_cr_ in the range of 10 to 60 mL/min are displayed separately by the minimum inhibitory concentrations (MICs) in Figure 3. It was almost difficult to achieve PTA ≥ 90% at a dosing interval of 12 h, despite the higher dose in every MIC level. In the MICs of 4 and 8 μg/mL, a 6-h dosing interval showed acceptable PTA regardless of the dose through 10–60 mL/min of CL_cr_, although PTA ≥ 90% was not achieved by 8-h interval regimens in the larger CL_cr_ levels. For MICs of more than 16 μg/mL and CL_cr_ > 60, no regimens led to an acceptable PTA.

Table 4 shows the PK/PD breakpoints for PIPC regimens in patient populations with various degrees of CL_cr_. For every patient population, the PK/PD breakpoint increased in the following order: 4.5 g every 6 h, > 2.25 g every 6 h, > 4.5 g every 8 h, > 2.25 g every 8 h, > 4.5 g every 12 h, and > 2.25 g every 12 h. The PTAs for TAZ at the PD target of area under the free-plasma-concentration–time curve over 24 h (*f*AUC_0−24_) ≥ 96 μg h/mL for the several regimens are shown in Figure 4. The regimens of 0.5 g every 8 h and 0.5 g every 6 h as TAZ doses achieved PTA ≥ 90% in all patients, whilst PTA ≥ 90% was not obtained by the regimens of 0.5 g every 12 h, 0.25 g every 6 h in CL_cr_ levels over 50 mL/min, or 0.25 g every 8 h in CL_cr_ levels over 30 mL/min. The regimen of 0.25 g every 12 h did not achieve PTA ≥ 90% in any CL_cr_ level.

## 3. Discussion

Some population PK studies concerning PIPC/TAZ in adult subjects have been reported [10,11,12,13,14,15,16,17,18,19]. In many antibiotics, including carbapenems and cephems, elderly patients have been reported to have different PK parameters than younger patients [21,22,23]. These age-related differences in PK parameters would increase the likelihood of drug toxicity and adverse reactions [24]. There are some reports that PK parameters and plasma concentrations were different between early and late elderly patients [25,26]. Therefore, in this study, we performed population PK analysis and PK/PD analysis for PIPC/TAZ only in late elderly patients.

The PK models of PIPC and TAZ have been previously described by both one- [16,18,19] and two-compartment models [10,12,13,14,15,17]. In our study, plasma-concentration–time data were analyzed by a two-compartment model because the AIC values of the two-compartment model (1040.069, PIPC and 596.955, TAZ) were lower than those of the one-compartment model (1063.038, PIPC and 615.059, TAZ). We collected venous blood samples at 0, 1, 1.5, 2, 3, and 5 h after the beginning of the administration. The presence of many blood sampling points in the distribution phase (<2 h) may be why our model fits better with the two-compartment model.

The present study identified CL_cr_ calculated with the Cockcroft–Gault equation as the most significant covariate that affected the PK in late elderly patients. The PK parameters obtained in this study were compared with those reported previously. To compare PK parameters, CL, V_c,_ and V_p_, obtained in this study with those reported previously, each parameter was quantified by substituting the population mean for the covariates. In this study, the CL values for late elderly patients estimated from the population parameters and mean CL_cr_ values were 4.62 and 5.04 L/h for PIPC and TAZ, respectively. These are lower than the values of PIPC (5.36 to 18.02 L/h) and TAZ (6.95 to 8.03 L/h) in adults including young and elderly patients, which was reported in previous papers [10,12,13,14,15,17]. This may be due to age-related changes in the elimination process, namely decreased renal function. Renal-excretion-type drugs are significantly affected by age [27,28]. Both PIPC and TAZ are known to be excreted primarily from the kidney, with the majority (70%) being eliminated as unchanged in urine [21,29]. On the other hand, Hamada et al. reported that the CL values for PIPC and TAZ in elderly Japanese patients with the age of 65 ± 17 years old were 8.54 and 9.03 L/h, respectively [19]. The CL values obtained by our analysis were still lower than these values. Therefore, it may be necessary to use unique parameters for patients over the age of 75, not as a whole, but as a special patient population.

As for the distribution volume of PIPC, V_c_ and V_p_ in this study were 5.39 and 6.96 L, respectively. In the previous reports, the ranges for V_c_ and V_p_ were 7.2–18.9 L and 3.5–17.8 L, respectively, where V_c_ was less than the reported value and V_p_ was within the reported range [10,12,13,14,15,17]. In TAZ, V_c_ and V_p_ in this study were 6.29 L and 7.73 L, respectively. In Korean patients with acute infections [15], TAZ had V_c_ and V_p_ of 21.7 L and 4.3 L, respectively. These results indicate that the distribution volume obtained in this study is particularly low in V_c_ compared to the previously reported values. In elderly people, the distribution volume of water-soluble drugs, such as PIPC and TAZ, decreases because the body water content of the elderly is lower than that of younger people. This may explain the small value of the distribution volume estimated in this study. In our model, the distribution volume had no covariates, while earlier reports have shown body weight as a covariate [13,15]. In those previous studies, young and obese patients were included and the distribution of body weight was broad. The lack of covariates such as body weight in this study may be related to the relatively narrow range of body weight of 32.0–68.7 kg, which is due to patients being elderly.

The goodness-of-fit plots are depicted in Figure 2. As observed in the plots with observed vs. population and individual predicted concentrations, the data are uniformly scattered around the line of unity. The conditional weighted residual plots show that concentrations can be predicted with less bias in the wide range of PIPC levels and time after dosing, as all residuals are uniformly scattered around zero.

In our PK/PD analysis of PIPC, the percentage of time that the unbound drug concentration exceeds the MIC for at least 50% of the dosing interval (50% *f*T > MIC) was used as the parameter of PTA assessment. As shown in Figure 3, random simulation demonstrated that PTA for 2.0 g (PIPC dose) every 6 h (8 g/day) was comparable to that for 4 g (PIPC dose) every 8 h (12 g/day). This suggests that the dosing interval is a more important factor than the daily dose in the dosing strategy for PIPC, which exhibits time-dependent killing. The guidelines use three CL_cr_ levels (<10, 10–50, and >50 mL/min) to evaluate renal function and recommend a different drug administration approach for each level [8,30]. As shown in this study, however, it is possible to recommend more detailed dosing regimens based on individual patient’s CL_cr_ levels and the MIC of PIPC.

For TAZ, there are several studies in which the %T > threshold value was used as the PD target [31]. However, Chung E.K et al. reported that it was difficult to achieve PTA > 90% for the usual administration method, when the PD target was %T > threshold value [16]. Therefore, in the present study, the PTA for TAZ was calculated for each dosing regimen using another PD target of *f*AUC_0-24_ ≥ 96 μg h/mL [32]. The results of PTA, shown in Figure 4, indicate that the recommended dosing regimens were 0.25 g every 8 h, 0.25 g every 6 h, 0.5 g every 12 h, and 0.5 g every 8 h for the CL_cr_ levels of 10, 20, 30–40, and 50–60 mL/min, respectively. In chemotherapy with PIPC/TAZ, it is necessary to consider the optimal dosing method of not only PIPC but also TAZ, a β-lactamase inhibitor, so that dosage regimens should be evaluated in combination with both PIPC and TAZ. The recommended administration methods based on our population PK/PD analysis of PIPC and TAZ are shown in Table 5. It has been shown that clinical MIC breakpoints for PIPC/TAZ against extended-spectrum beta-lactamase (ESBL)-producing Enterobacteriaceae are 8/4 μg/mL in Japan [33]. For example, the patients with CL_cr_ of 50–60 mL/min and ESBL-producing Enterobacteriaceae (MIC = 8 μg/mL) are recommended PIPC/TAZ 4.5 g every 6 h in this study. In this case, the same regimen is recommended in the Sanford Guide [8]. In the patients with CL_cr_ of 50–60 mL/min and a MIC ≤ 4 μg/mL, the recommended dosage in Table 5 is PIPC/TAZ 4.5 g every 8 h. On the other hand, the Sanford Guide recommends the same regimen for patients with CL_cr_ > 40 mL/min regardless of MIC. Thus, using Table 5 prepared by this study, we were able to present the optimal dosing regimen for the elderly in the late stage that takes into account not only CL_cr_ but also the MIC.

Additionally, we have already reported that renal dysfunction, especially CL_cr_ of less than 40 mL/min, could be an influential factor for nephrotoxicity induced by PIPC/TAZ-induced nephrotoxicity in late elderly patients [34]. Therefore, although the recommended dosing regimens are shown in Table 5, from the viewpoint of safety, the administration of PIPC/TAZ should be performed more carefully in patients with CL_cr_ values of 30 mL/min or less.

As mentioned above, it was possible to estimate PTA in various dosing methods of PIPC/TAZ corresponding to various MICs according to renal function and to show recommended administration schedules. This is the first report showing a PK/PD analysis-based treatment planning for PIPC/TAZ therapy in late elderly patients, where careful treatment planning should be considered.

This study has some limitations. First, the number of patients used in the analysis was too small to be conclusive. Second, since TAZ is currently only clinically available as a combination preparation, TAZ concentration data were obtained in the presence of PIPC. This model may not be applicable to PIPC or TAZ alone, as it has been reported that the PK of TAZ changes when coadministered with PIPC [35].

There are some problems in pharmacotherapy for elderly people, as (1) they often have multiple chronic diseases [36], (2) they have twice or three times as many adverse drug reactions as young adults [37,38], and (3) the clinical practice guidelines are not sufficiently established [39]. Therefore, our study in late elderly people may be helpful for the treatment of infectious disease in super-aged society worldwide. Moreover, the implications of our findings and proposals should be verified in the clinical setting.

## 4. Materials and Methods

### 4.1. Patients and Ethics

Subjects were inpatients over 75 years old (to be late elderly patients) with pneumonia at the Department of Clinical Oncology and Respiratory Medicine, Shimane University Hospital. The cases falling under the following criteria were excluded: (1) cases with serious heart/liver/renal function failure, (2) cases in which atypical pneumonia is strongly suspected, (3) cases with a history of allergies to β-lactam antibiotics.

The research was conducted in accordance with the Declaration of Helsinki and national and institutional standards. This study was approved by the Ethical Committee of Shimane University Faculty of Medicine (No.742 and 743). All patients or their legal representatives provided written informed consent prior to study enrollment.

### 4.2. Drug Administration, Sampling Procedure, and Analysis

PIPC/TAZ (4.0 g/0.5 g) was dissolved in 100 mL saline, and intravenous drip infusion was carried out for 1 h each session 3 times a day. In patients with the estimated glomerular filtration rate (eGFR) of < 50 mL/min, the dose was reduced to 2.25 g of PIPC/TAZ (2.0 g/0.25 g) each session 3 times a day in accordance with the Sanford Guide to Antimicrobial Therapy (the Sanford Guide) [8].

Venous blood samples (3 mL) were collected in heparinized tubes at (0) and 1, 1.5, 2, 3, and 5 h after the beginning of the administration. Plasma was immediately separated from the blood samples by centrifugation at 3000 rotations/m for 10 min at 4 °C and stored at −30 °C until analysis. The plasma total concentrations of PIPC and TAZ were simultaneously measured by using high-performance liquid chromatography with UV detection in accordance with the method reported previously [40]. The quantification range was 0.5–1000 μg/mL for both PIPC and TAZ concentrations in plasma samples. For intra- and interday measurement, the precision was 0.69–7.24% (PIPC) and 1.10–9.49% (TAZ), and the accuracy was 99.6–107% (PIPC) and 97.5–112% (TAZ). No peak interfering with the quantification was detected on the chromatograms of the measurement samples.

### 4.3. Population Pharmacokinetic (PK) Analysis

Population PK analysis was performed using the first-order conditional estimation (FOCE) method in the NONMEM program (version 7.3.0; ICON Development Solutions, Ellicott City, MD, USA). Both one- and two-compartment models were fitted to the data when choosing a base PK model. Interindividual variability was modeled exponentially: *θ*_i_ = *θ* × exp(η_i_), where *θ*_i_ is the fixed-effects parameter for the ith subject, *θ* is the mean value of the fixed-effects parameter in the population, and η is a random interindividual variable which is normally distributed with a mean of zero and a variance of ω^2^. The residual (intraindividual) variability was modeled with a combined additive-proportional error model: C_obs,ij_ = C_pred,ij_ × (1 + ε_proprtional,ij_) + ε_additive,ij_, where C_obs,ij_ and C_pred,ij_ are the jth observed and predicted concentrations for the ith subject, and ε is a random intraindividual error which is normally distributed with a mean of zero and a variance of σ^2^. Model discrimination was assessed using AIC values. Parameter uncertainty, presented as 95% confidence intervals (95% CI), was calculated with the sampling importance resampling procedure, as recently described by Dosne et al. [41].

The influences of the patient characteristics on the individual PK parameters obtained from the base model were explored graphically. The covariates showing a correlation with the PK parameters were introduced into the model. The significance of the influence of the covariates was evaluated by the changes of −2 log-likelihood (the minimum value of the objective function: OBJ). An OBJ decrease of more than 3.84 from the base model (*p* < 0.05; χ^2^ test) was considered statistically significant during the forward inclusion process. The full model was built by incorporating the significant covariates, and the final model was developed by a backward elimination method. The covariates in the full model were excluded from the model one at a time, and an OBJ increase of more than 6.63 from the full model (*p* < 0.01; χ^2^ test) was considered statistically significant.

The validity of the final model was evaluated by the bootstrap method (1000 replications) using the Perl-speaks-NONMEM program, version 3.7.6 (Karlsson M, Hooker A, Nordgren R, Harling K; Uppsala University, Uppsala, Sweden). Goodness-of-fit included the visual inspection of the following plots: observed vs. population predicted concentrations; observed vs. individual predicted concentrations; conditional weighted residuals vs. time; and conditional weighted residuals vs. population predicted concentrations.

### 4.4. PK/PD Analysis Using a Random Simulation

A random simulation, often called a Monte Carlo simulation, was conducted using the final population PK model to estimate the PTA profile for various PIPC/TAZ regimen (1-h infusion)-MIC combinations. The simulation process was repeated 1000 times as follows: a set of fixed-effects parameters was generated randomly according to each mean estimate and interindividual variance of the final population PK model. The steady-state unbound-drug-concentration–time curve was simulated using the fixed-effects parameters, where a value of 30% protein binding in humans was employed [29].

For β–lactam antibiotics, the PK/PD index that best links drug exposure with the antibacterial effect is the fraction of time that the free drug concentrations are above the MIC. For PIPC, the percentage of time that the unbound drug concentration exceeds the MIC is at least 50% of the dosing interval (50% *f*T > MIC), which is generally associated with its maximal efficacy [42]. Therefore, we used 50% *f*T > MIC as a parameter in our study. The time point at which the free drug concentration coincided with a specific MIC value was determined, and the time for which the *f*T > MIC was finally calculated as the cumulative percentage of a 24-h period. The PTA (%) was determined as the fraction that achieved at least 50% *f*T > MIC (the bactericidal target) of 1000 estimates [43].

The PTA for TAZ was calculated for each dosing regimen using the PD target of *f*AUC_0−24_ ≥ 96 μg h/mL [32]. *f*AUC_0−24_ was calculated for each of the 1000 simulated patients for each dosing regimen. PTA was then calculated by dividing the number of patients achieving *f*AUC_0−24_ ≥ 96 μg h/mL by the total number of simulated patients. The *f*AUC has been shown to be an important PK parameter for the in vitro activity of TAZ, and the *f*AUC_0−24_ of 96 μg h/mL was derived from *f*AUC of TAZ during in vitro susceptibility testing (4 μg/mL × 24-h incubation = 96 μg h/mL).

## 5. Conclusions

In this study, a population PK model of PIPC/TAZ for elderly people over 75 years of age was indicated. PK/PD analysis using population parameters and MIC values enabled the presentation of optimal regimens based on renal function and the MIC of the bacteria. In the future, it will be necessary to verify the clinical usefulness of the dosing regimen recommendation for PIPC/TAZ therapy by the present method.

## Figures and Tables

**Figure 1 antibiotics-09-00113-f001:**
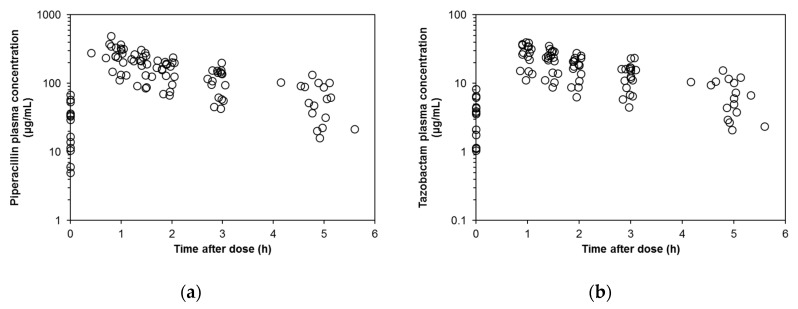
Piperacillin- and tazobactam-concentration–time data (100 samples each). (**a**) Piperacillin; (**b**) Tazobactam.

**Figure 2 antibiotics-09-00113-f002:**
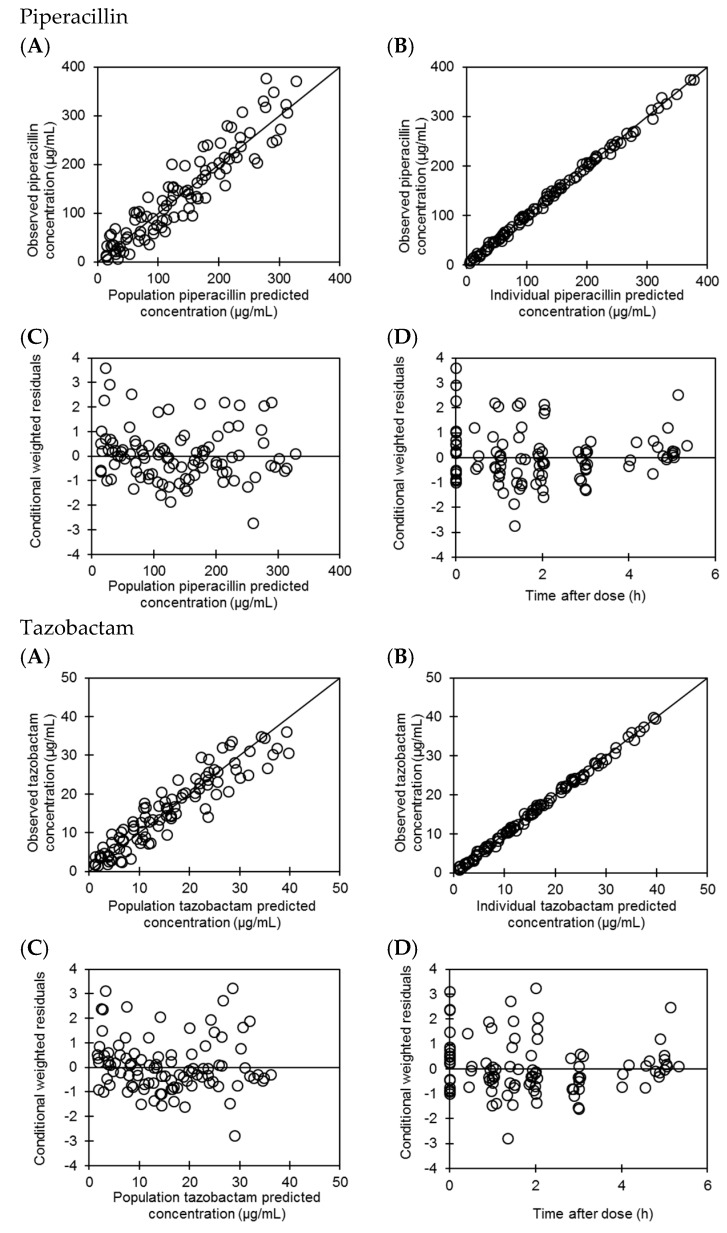
Goodness-of-fit plot of the final model. (**A**) Observed vs. population predicted concentration, (**B**) Observed vs. individual predicted concentration, (**C**) Conditional weighted residuals vs. population predicted concentration, (**D**) Conditional weighted residuals vs. time after dose.

**Figure 3 antibiotics-09-00113-f003:**
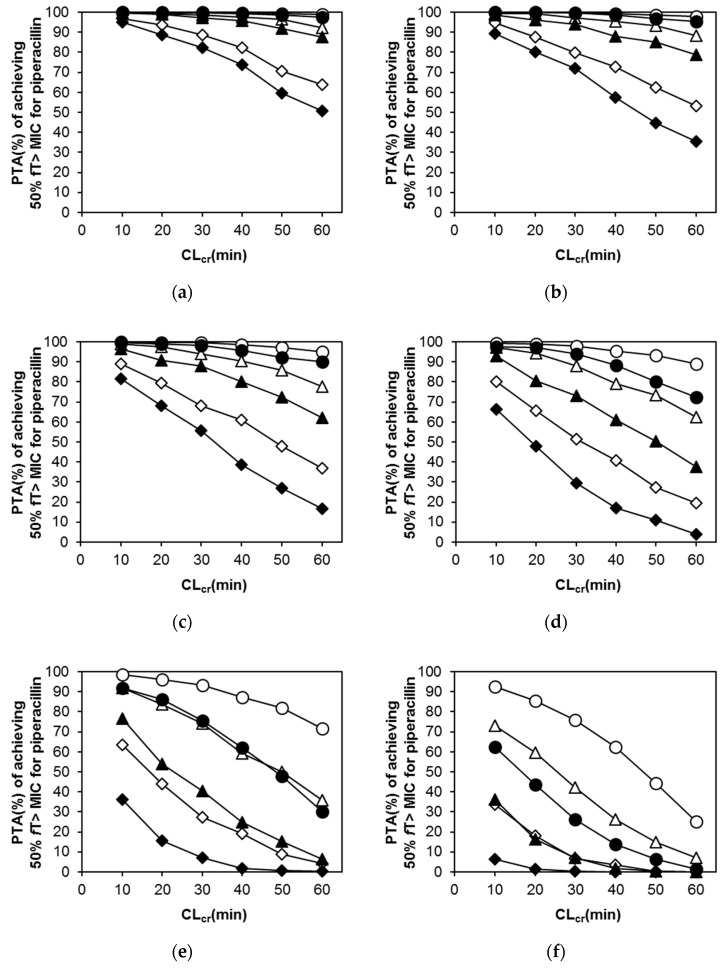
Probability of target attainment (PTA) of achieving 50% *f*T > MIC for piperacillin when administered in late elderly patients. MIC: minimum inhibitory concentration, CL_cr_: creatinine clearance. (**a**) MIC = 2 μg/mL; (**b**) MIC = 4 μg/mL; (**c**) MIC = 8 μg/mL; (**d**) MIC = 16 μg/mL; (**e**) MIC = 32 μg/mL; (**f**) MIC = 64 μg/mL. **○:** 4.5 g(PIPC4.0 g) q6h, △: 4.5 g(PIPC4.0 g) q8h, ◇: 4.5 g (PIPC4.0 g) q12h, ●: 2.25 g (PIPC2.0 g) q6h, ▲: 2.25 g (PIPC2.0 g) q8h, ◆: 2.25 g(PIPC2.0 g) q12h.

**Figure 4 antibiotics-09-00113-f004:**
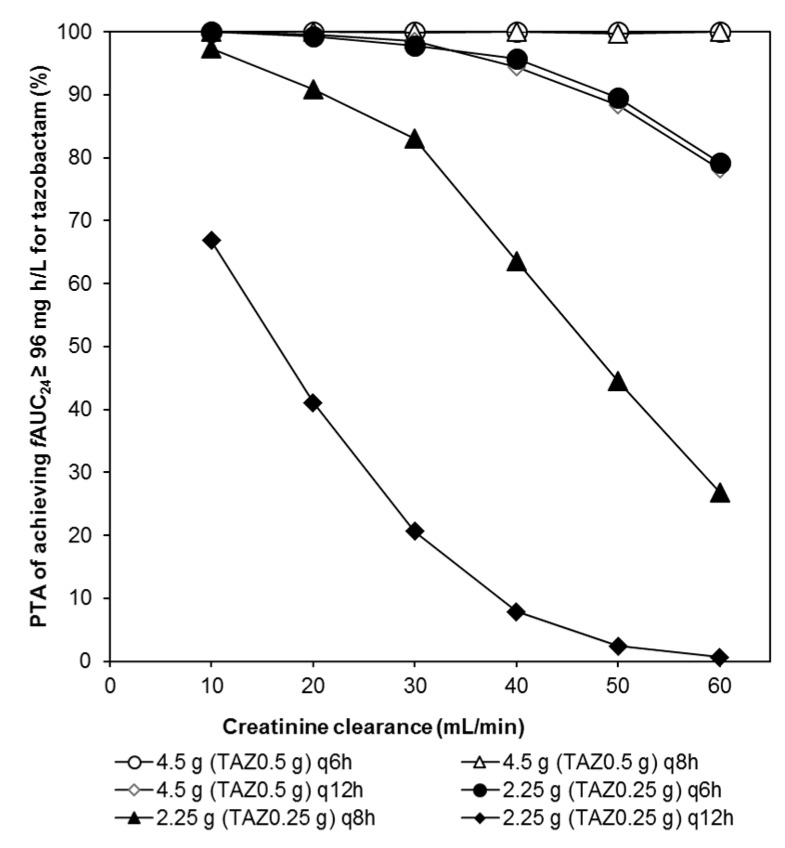
Probability of target attainment (PTA) of achieving *f*AUC_0−24_ ≥ 96 mg h/L for tazobactam when administered in late elderly patients. *f*AUC_0−24_, area under the free-plasma-concentration–time curve over 24 h.

**Table 1 antibiotics-09-00113-t001:** Demographic patient data.

Subjects Number (Male/Female)	18 (14/4)
Characteristic	Mean ± S.D. (Range)
Age (years)	86.5 ± 6.0 (75–101)
Height (cm)	154.1 ± 7.8 (138.0–165.0)
Weight (kg)	45.5 ± 10.0 (32.0–68.7)
Body mass index	19.1 ± 3.5 (13.9–27.3)
Serum creatinine (mg/dL)	0.91 ± 0.31 (0.60–1.55)
Creatinine clearance (mL/min) ^(a)^	38.0 ± 11.1 (21.5–59.1)
Serum albumin (g/dL)	2.9 ± 0.6 (2.1–3.7)

^(a)^ Cleatinine clearance (CL_cr_) was estimated by the Cockcroft-Gault formula.

**Table 2 antibiotics-09-00113-t002:** Estimate of population pharmacokinetic (PK) parameters of piperacillin (PIPC) in the final model.

Parameter	Population Estimate	Standard Error	95% Confidence Interval (Bootstrap Procedure)
**Fixed-effects parameter**			
**CL (L/h) = *θ*_1_ + *θ*_2_ × (CL_cr_ − 37.4)**			
***θ*_1_**	4.58	0.289	4.04–5.22
***θ*_2_**	0.061	0.0293	0.0107–0.114
**V_c_ (L) = *θ*_3_**			
***θ*_3_**	5.39	0.969	3.81–8.33
**Q (L/h) = *θ*_4_**			
***θ*_4_**	20.7	6.11	12.0–37.1
**V_p_ (L) = *θ*_5_**			
***θ*_5_**	6.96	0.314	5.09–7.84
**Interindividual variability**			
ω^2^_CL_	0.0705 (CV = 27.0%)	0.0177	0.0357–0.104
ω^2^_Vc_	0.389 (CV = 69.0%)	0.153	0.118–0.803
ω^2^_Q_	0.311 (CV = 60.4%)	0.327	0.00990–1.40
ω^2^_Vp_	0 (fixed)	None	None
**Residual variability**			
σ^2^_proprtional_	0.000927	0.000798	0.000170–0.00289
σ^2^_additive_	25.1	12.8	3.72–46.6

Estimates are expressed as means. CL: clearance, CL_cr_: creatinine clearance (mL/min), V_c_: central volume of distribution, Q; intercompartment clearance, V_p_: peripheral volume of distribution. CV: coefficient of variation. The median value of creatinine clearance was 37.4.

**Table 3 antibiotics-09-00113-t003:** Estimate of population PK parameters of tazobactam (TAZ) in the final model.

Parameter	Population Estimate	Standard Error	95% Confidence Interval (Bootstrap Procedure)
**Fixed-effects parameter**			
**CL (L/h) = *θ*_1_ + *θ*_2_ × (CL_cr_ – 37.4)**			
***θ*_1_**	5.00	0.318	4.41–5.78
***θ*_2_**	0.0587	0.0298	0.0116–0.125
**V_c_ (L) = *θ*_3_**			
***θ*_3_**	6.29	1.04	4.13–11.1
**Q (L/h) = *θ*_4_**			
***θ*_4_**	24.0	8.44	9.90–44.6
**V_p_ (L) = *θ*_5_**			
***θ*_5_**	7.73	0.443	5.61–8.31
**Interindividual variability**			
ω^2^_CL_	0.0715 (CV = 27.2%)	0.0221	0.0301–0.125
ω^2^_Vc_	0.547 (CV = 85.3%)	0.244	0.169–1.11
ω^2^_Q_	0.545 (CV = 85.1%)	0.465	0.0340–1.79
ω^2^_Vp_	0 (fixed)	None	None
**Residual variability**			
σ^2^_proprtional_	0.000479	0.000749	0.0000499–0.00252
σ^2^_additive_	0.394	0.223	0.0297–0.602

Estimates are expressed as means. CL: clearance, CL_cr_: creatinine clearance (mL/min), V_c_: central volume of distribution, Q; intercompartment clearance, V_p_: peripheral volume of distribution. CV: coefficient of variation. The median value of creatinine clearance was 37.4.

**Table 4 antibiotics-09-00113-t004:** Pharmacokinetic/pharmacodynamic breakpoints for piperacillin regimens in patient populations with various degrees of creatinine clearance.

Piperacillin/Tazobactam Regimen	Creatinine Clearance (mL/min)					
	60	50	40	30	20	10
**4.5 g q6h**	8 μg/mL	16 μg/mL	16 μg/mL	32 μg/mL	32 μg/mL	64 μg/mL
**4.5 g q8h**	2 μg/mL	4 μg/mL	8 μg/mL	8 μg/mL	16 μg/mL	32 μg/mL
**4.5 g q12h**	0.125 μg/mL	0.25 μg/mL	0.5 μg/mL	1 μg/mL	2 μg/mL	4 μg/mL
**2.25 g q6h**	8 μg/mL	8 μg/mL	16 μg/mL	16 μg/mL	16 μg/mL	32 μg/mL
**2.25 g q8h**	1 μg/mL	2 μg/mL	8 μg/mL	8 μg/mL	8 μg/mL	16 μg/mL
**2.25 g q12h**	0.063 μg/mL	0.125 μg/mL	0.25 μg/mL	0.5 μg/mL	1 μg/mL	2 μg/mL

Defined as the highest minimum inhibitory concentration (MIC) at which the probability of 50% *f*T > MIC attainment in plasma was ≥90%.

**Table 5 antibiotics-09-00113-t005:** Optimal dosing regimens of piperacillin/tazobactam in elderly patients over 75 years old.

MIC (μg/mL)	Creatinine Clearance (mL/min)					
	60	50	40	30	20	10
**2**	4.5 g q8h	4.5 g q8h	2.25 g q6h	2.25 g q6h *	2.25 g q8h *	2.25 g q8h *
**4**	4.5 g q8h	4.5 g q8h	2.25 g q6h	2.25 g q6h *	2.25 g q8h *	2.25 g q8h *
**8**	4.5 g q6h	4.5 g q6h	2.25 g q6h	2.25 g q6h *	2.25 g q8h *	2.25 g q8h *
**16**	-	4.5 g q6h	4.5 g q6h	2.25 g q6h *	2.25 g q6h *	2.25 g q8h *
**32**	-	-	-	4.5 g q6h *	4.5 g q6h *	2.25 g q6h *
**64**	-	-	-	-	-	4.5 g q6h *

MIC: minimum inhibitory concentration. *: Renal impairment, especially creatinine clearance of < 40 mL/min, requires attention to the nephrotoxicity induced by piperacillin/tazobactam administration. -: Not recommended.

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
