# Peer review of "Population Pharmacokinetic Modeling and Pharmacodynamic Target Attainment Simulation of Piperacillin/Tazobactam for Dosing Optimization in Late Elderly Patients with Pneumonia"

_antibiotics, 2020, doi:10.3390/antibiotics9030113_

Round 1

Reviewer 1 Report

Manuscript entitled "Population pharmacokinetic modeling and pharmacodynamic simulation of piperacillin/tazobactam for dosing optimization in late elderly patients with pneumonia" authors successfully demonstrated the advantage of PBPK modeling in designing personalized medicine for elderly patients undergoing antibiotics. However, there are missing links which needs to be addressed before the manuscript can be deemed publishable.

1) Please improve on the introduction part. It does not look like in its current state. Please provide more facts.

2) How did the authors perform compartment modeling with just few sampling points (0-4 h). Please discuss the rationale why few sampling points are considered and why authors think that this does not impart any bias on the outcome of the study.

3) Analytical method details are missing in the manuscript. What is the calibration range? How do authors justify the HPLC-UV analysis results, as it is not a specific detection method? Is the method validated? Please furnish chromatograms and a brief summary of the analytical method.

Overall, nice piece off work. Furnishing above details in the manuscript would make it more interesting to the readers.

Reviewer 2 Report

See PDF.
